# Resveratrol and Quercetin as Regulators of Inflammatory and Purinergic Receptors to Attenuate Liver Damage Associated to Metabolic Syndrome

**DOI:** 10.3390/ijms22168939

**Published:** 2021-08-19

**Authors:** Agustina Cano-Martínez, Rocío Bautista-Pérez, Vicente Castrejón-Téllez, Elizabeth Carreón-Torres, Israel Pérez-Torres, Eulises Díaz-Díaz, Javier Flores-Estrada, Verónica Guarner-Lans, María Esther Rubio-Ruíz

**Affiliations:** 1Department of Physiology, Instituto Nacional de Cardiología Ignacio Chávez, Juan Badiano 1, Sección XVI, Tlalpan, Mexico City 14080, Mexico; agustina.cano@cardiologia.org.mx (A.C.-M.); vicente.castrejon@cardiologia.org.mx (V.C.-T.); veronica.guarner@cardiologia.org.mx (V.G.-L.); 2Department of Molecular Biology, Instituto Nacional de Cardología Ignacio Chávez, Juan Badiano 1, Sección XVI, Tlalpan, Mexico City 14080, Mexico; rociobtst@yahoo.com (R.B.-P.); juana.carreon@cardiologia.org.mx (E.C.-T.); 3Department of Cardiovascular Biomedicine, Instituto Nacional de Cardiología Ignacio Chávez, Juan Badiano 1, Sección XVI, Tlalpan, Mexico City 14080, Mexico; israel.perez@cardiologia.org.mx; 4Department of Reproductive Biology, Instituto Nacional de Ciencias Médicas y Nutrición “Salvador Zubirán”, Vasco de Quiroga 15, Sección XVI, Tlalpan, Mexico City 14080, Mexico; eulisesd@yahoo.com; 5División de Investigación, Hospital Juárez de México, Av. Instituto Politécnico Nacional 5160, Magdalena de las Salinas, Gustavo A. Madero, Mexico City 07760, Mexico; jose.florese@salud.gob.mx

**Keywords:** inflammation, liver damage, toll-like receptor 4, P2Y2 receptor, metabolic syndrome, resveratrol, quercetin

## Abstract

Nonalcoholic fatty liver disease (NAFLD) is considered a manifestation of metabolic syndrome (MS) and is characterized by the accumulation of triglycerides and a varying degree of hepatic injury, inflammation, and repair. Moreover, peroxisome-proliferator-activated receptors (PPARs) play a critical role in the pathophysiological processes in the liver. There is extensive evidence of the beneficial effect of polyphenols such as resveratrol (RSV) and quercetin (QRC) on the treatment of liver pathology; however, the mechanisms underlying their beneficial effects have not been fully elucidated. In this work, we show that the mechanisms underlying the beneficial effects of RSV and QRC against inflammation in liver damage in our MS model are due to the activation of novel pathways which have not been previously described such as the downregulation of the expression of toll-like receptor 4 (TLR4), neutrophil elastase (NE) and purinergic receptor P2Y2. This downregulation leads to a decrease in apoptosis and hepatic fibrosis with no changes in hepatocyte proliferation. In addition, PPAR alpha and gamma expression were altered in MS but their expression was not affected by the treatment with the natural compounds. The improvement of liver damage by the administration of polyphenols was reflected in the normalization of serum transaminase activities.

## 1. Introduction

An increase in the intake of sugars (sucrose and fructose), a lack of physical activity, and genetic predisposition predict the development of metabolic syndrome (MS), independently from obesity and the prevalence of this disease is increasing dramatically in Western and developing countries [1]. MS is a cluster of cardiovascular risk factors associated with obesity and insulin resistance (IR) and is strongly linked to an increase in the level of systemic inflammation markers such as C-reactive protein (CRP), interleukin 6 (IL-6), and tumor necrosis factor-alpha (TNF-α) and an increase in the free fatty acid (FFA) concentration [2]. This disorder is not only associated with a higher risk of appearance of type 2 diabetes and cardiovascular events, but it also impacts the liver in different ways. Nonalcoholic fatty liver disease (NAFLD) is considered the hepatic manifestation of MS and is characterized by triglyceride accumulation and a variable degree of hepatic injury, inflammation, and repair [3,4]. Moreover, some reports suggest a link between liver inflammation and IR [3,4,5].

FFA, such as saturated fatty acids, activate toll-like receptors (TLR), which are a family of surface receptors that are present in all cells and are typically involved in the innate immune responses [2]. Particularly, TLR2 and TLR4 play a key role in obesity-related inflammation, IR, and vascular dysfunction [6,7]. In the liver, TLR expression was observed on a variety of cells and plays an important part in multiple liver diseases [8]. However, the direct role of TLR4 in these processes in the liver tissue is unclear [9].

When exposed to inflammatory stimuli, neutrophils release a large group of serine proteases, among which neutrophil elastase (NE) is the most important [10]. Obesity is associated with an increase in NE activity and NE is also implicated in IR by inhibiting hepatic Insulin receptor substrate 1. This effect is dependent on the activation of TLR4 [11,12].

After an inflammatory signal, adenosine 5′-triphosphate (ATP) is released into the extracellular space. ATP is known for its important role in intracellular cell metabolic pathways; however, this nucleotide can also act as a danger signal on the purinergic receptors (P2X or P2Y) which are diffusely expressed in various organs including the liver [13]. These receptors are essential regulators of physiological functions and serve as danger signals that trigger inflammation after injury [14]. Purinergic signaling, by P2Y, interacts with other signaling molecules to form a complex network, regulating numerous cellular processes including phagocytosis, chemotaxis, cytokine production, proliferation, differentiation, and death [13,15]. P2Y2 receptors are also associated with fat accumulation, hepatic steatosis, IR, metabolic complications, and inflammation [16,17].

On another hand, peroxisome-proliferator-activated receptors (PPARs) are activated through endogenous agonists (fatty acids and their derivatives) or exogenous agonists and regulate transcriptional activity [18]. Each PPAR isotype possesses specific functional characteristics to control a whole spectrum of physiological functions in the liver, including oxidative stress, lipid, and glucose metabolism, inflammatory responses, regenerative mechanisms, and cell differentiation and proliferation [19].

PPARγ is generally increased in livers with steatosis of both animal models of obesity and NAFLD patients. As opposed to PPARγ, PPARα plays a critical role in the regulation of fatty acid uptake, beta-oxidation, ketogenesis, synthesis of bile acid, and turnover of triglycerides to prevent hepatic steatosis [18]. In addition to its role in the regulation of metabolism, PPARα also has anti-inflammatory effects by inhibiting TLR4 expression and by inhibiting the NF-κB signaling pathway [20]. PPARγ has emerged as a potential target for the treatment of inflammatory diseases such as ulcerative colitis, atherosclerosis, asthma, and rheumatoid arthritis [21]. However, as far as we know, there are no reports on the association of PPAR expression and P2Y2 protein levels in the liver.

In recent years polyphenols such as resveratrol (RSV) and quercetin (QRC), which are present in fruits and vegetables, have gained interest by researchers for preventing and treating diseases, including obesity and obesity-related metabolic diseases [22]. These molecules are available as pills or capsules and people take these nutritional supplements. Although there are studies that demonstrate the antihypertensive, antioxidant, and anti-inflammatory properties in different human and animal models, the mechanisms underlying the beneficial effects of RSV and QRC have not been fully elucidated [22,23,24]. 

There is also little evidence of the effect of polyphenols on liver disorders associated with inflammatory and metabolic signaling through TLR4 and purinergic receptors [25]. Although some authors have identified flavonoid derivatives such as potent P2Y2 receptor antagonists, little is known about the effect of flavonoids on purinergic receptor expression [26,27]. Hence, the present study aimed to evaluate the effect of RSV and QRC mixture on the expression of TLR4, NE, and P2Y2 receptors and their association with the expression of PPARs. In addition, we assessed whether the expression of these elements was associated with fibrosis, apoptosis, and proliferation in an MS rat model.

## 2. Results

### 2.1. Metabolic Syndrome 

The characterization of the MS model was done by analyzing the animal body weight, blood pressure, and intra-abdominal fat and by the serum biochemical analysis. As shown in Table 1, MS animals had an increased body weight and they developed central obesity, hypertension, dyslipidemia (high levels of triglycerides), hyperinsulinemia, and IR (HOMA-IR). Serum adipokine concentrations are higher in the MS group when compared to the Control group.

As expected, the treatment with RSV + QRC significantly decreased body weight, central adiposity, hypertension, hypertriglyceridemia, and restored IR in the MS group. In the Control group, polyphenol-administration did not affect the body or serum parameters.

### 2.2. TLR4 Expression

Figure 1 shows the expression of TLR4 in the liver from Control and MS rats. The presence of a label for TLR4 was located in hepatocytes around the central vein. The proportion of TLR4 in the MS group was 2.8 times higher compared to the Control. RSV + QRC administration significantly diminished TLR4 expression in both Control and MS groups although this effect was more evident in the MS rats (53% vs. 87%, respectively) (Figure 1C,D).

### 2.3. Neutrophil Elastase (NE) Expression

Due to the association of NE with TRL4 expression, we studied if the administration of natural compounds exerts an effect on this enzyme. The presence of the label for NE was located in regions away from the lumen of the vessels. The proportion of NE located in the liver of rats with MS was 5 times higher than that detected in the Control (Figure 2A,B). RSV + QRC treatment in Control and MS animals reverses the proportion of NE by 80% (Figure 2C,D).

### 2.4. P2Y2 Expression

Figure 3 revealed the differences in the expression of P2Y2 in livers from the Control and MS groups. The presence of labels for P2Y2 receptors was located mainly in hepatocytes around the central vein. The proportion of P2Y2 in the MS group was 32% higher compared to the Control. However, the treatment with natural compounds significantly diminished the P2Y2 expression in the same proportion (50% approximately) in both, Control and MS animals (Figure 3C,D).

### 2.5. Fibrosis

Because fibrosis is considered an indicator of liver damage, we analyze this parameter in the liver from the experimental groups. The liver of rats with MS presented mainly perivascular fibrosis (PVF), with indications of interstitial fibrosis (IF) and replacement fibrosis (RF) in the region surrounding the vessels, including both the central vein (CV) and the intralobular vein (ILBV) in the triad. Fibrosis was increased in the tissues from MS animals that were damaged similarly as was found with Masson’s trichrome staining (MT) (Figure 4) for total collagen deposits and confirmed with Sirius Red (SR) staining (Figure 5) for collagen I y III. The proportion of total collagen deposits in MS was 3 times more than that observed in the Control tissue. The RSV + QRC administration reduced deposition of fibrosis in the MS group almost reaching Control values (Figure 4).

Collagen I and III accumulation was confirmed by SR analysis. Livers from MS rats had 122% more collagen deposition compared to the livers from Control rats (Figure 5A,B). When Control and MS animals were treated with RSV plus QRC, both groups presented less collagen I and III depositions, although the decrease is more evident in MS animals (59% vs. 85%, respectively) (Figure 5C,D).

### 2.6. Apoptosis and Proliferation

Afterward, we researched if the treatment with polyphenols was able to prevent apoptosis using the Terminal deoxynucleotidyl transferase dUTP nick end labeling (TUNEL) assay (Figure 6). Our results showed that livers from rats with MS presents 3.34 times more cells in apoptosis compared to the Control animals (Figure 6A,B). The cells in apoptosis were located towards the lumen of the vessels as well as in the hepatocytes around the vessels, mainly in the central vein. RSV plus QRC treatment significantly reduced apoptosis (72%) in livers from MS rats; while the percentage did not change significantly in the Control group (Figure 6C,D).

The results shown in Figure 7 show the proliferative activity in liver tissue sections. Proliferating cell nuclear antigen (PCNA)-positive cells (brown color with a fine granular appearance) in the MS group were higher when compared with the Control group, and were localized as a part of infiltration. Our observations suggest that these cells could be Kupffer cells due to their localization in sinusoids (Figure 7A,B). The oral treatment with natural compounds significantly decreased the staining levels in the MS group; however, there were no significant differences in PCNA staining levels in the Control group.

### 2.7. Expression of PPARs 

We also evaluated the expression of PPARs isotypes in the experimental groups because they play a major role in metabolism and the inflammation process in the liver. Western blot analyses revealed differences in the expression of PPAR-α and PPAR-γ in liver homogenates from all groups (Figure 8). As expected, PPAR-α expression was higher in Control rats when compared to MS rats and PPAR-γ was increased in liver form MS rats. Nevertheless, protein levels of both PPARs isotypes were not modified by the treatment with natural compounds in both, Control and MS groups.

### 2.8. Activity of Transaminases

Finally, serum transaminases activity was determined in all groups because liver disease is often reflected by biochemical abnormalities. ALT and ALP activity was significantly higher in the MS than in Control rats (Table 2). This indicated liver damage in MS rats; however, no significant difference was seen between Control and MS groups in AST and GGT activities. Treatment with natural compounds significantly reduced ALT and ALP activities in MS animals (66% and 32%, respectively).

## 3. Discussion

Extensive evidence has demonstrated the beneficial effect of polyphenols on the treatment of cardiometabolic disorders. The mechanisms underlying the beneficial effects of RSV and QRC have not yet been fully elucidated and have mainly been related to epigenetic processes, intrinsic antioxidant activity, and anti-inflammatory mechanisms [23,24,28]. However, to the best of our knowledge, there are not reports on the effect of RSV and QRC on the pathways analyzed in this paper. In this work, we show that the decrease in TLR4 and NE expression are new mechanisms through which these natural compounds may have a protective role on liver damage associated with MS. These decreases are then followed by the diminution in liver fibrosis and apoptosis. Also, a novel finding presented in this study is that the treatment with RSV and QRC mixture is associated with the decrease in the expression of the P2Y2 receptor. Moreover, we evaluated if these anti-inflammatory effects are associated with differences in PPAR α and γ expression. 

Table 1 shows that the administration of RSV + QRC reversed some signs of MS such as body weight, central adiposity, hypertension, IR, and dyslipidemia without affecting the concentration of glucose, total cholesterol, adiponectin, and leptin; these results are in accordance with our previous reports [22,24,29]. 

Numerous studies have shown that different mechanisms exert synergic action in the development or progression of liver disease linked to MS such as accumulation of fatty acids, oxidative stress, and inflammation [24,28]. On this aspect, some authors have reported that TLR4 is involved in obesity and liver damage [7,30]. We analyzed the expression of the TLR4 receptor in livers from all experimental groups (Figure 1). Our results on the anti-inflammatory effects of polyphenols by decreasing TLR4 expression and its signaling pathway are in line with those reported by other authors [31,32]. Zhang [33] reported that non-esterified fatty acids (NEFAs), a crucial source of energy in the liver, may activate TLR4. On this aspect, in a previous study from our group, we found that circulating levels of NEFAs were higher in MS rats and that the RSV + QRC treatment reduces these levels [22]. This effect could be added to the decrease of TLR4 caused by FFA as a result of the presence of polyphenols since RSV plus QRC reduce the amount of FFA by increasing their oxidation in the liver [34]. Therefore, further studies are needed to support our hypothesis.

Another mechanism that mediates liver damage controlled by TLR4 is the activation of neutrophils and the release of NE. Although this enzyme might be a potential target to treat liver disorders linked to obesity and MS, there are still few studies reporting the effect of polyphenols on NE expression [11,13,35]. We observed that the livers from MS rats had an increase in this enzyme compared to livers from Control rats and that the RSV plus QRC mixture was able to abolish the expression of NE; however, it would be interesting to evaluate the effect of polyphenols treatment on NE activity.

Some authors have proposed that ATP serves as a messenger that links inflammation and metabolic derangements through its binding to the P2Y2 receptor [13,14,17]. Moreover, some reports have shown the therapeutic role of polyphenols by acting as antagonists of the P2Y2 receptors [25,26,27]. A novel finding of the present study is that the treatment with RSV plus QRC can decrease the expression of P2Y2 in livers from MS rats; hence, we suggest that this is a new mechanism for the therapeutic role of polyphenols in liver disease without ruling out their other pleiotropic effects.

Liver inflammatory and purinergic signaling modulate several physio-pathological processes such as proliferation, differentiation, migration, and death in response to injury [36,37]. Therefore, we analyzed the effect of the administration of polyphenols on the levels of fibrosis, apoptosis, and proliferation in livers from Control and MS animals (Figure 4, Figure 5, Figure 6 and Figure 7). Figure 4 and Figure 5 show that MS is associated with an increase in liver fibrosis compared to the Control group, and the administration of RSV + QRC mixture reverted this effect. Our results are in line with experimental and clinical evidence which suggests that RSV and QRC attenuate liver inflammation and fibrosis [28,38,39].

Results in Figure 6 demonstrate that livers from MS animals showed significantly higher levels of hepatocytes in apoptosis compared to the Control group. This effect was abolished by the treatment with polyphenols (Figure 6C,D). Our data are in accordance with previous studies that showed the association of apoptosis and liver disease as well as the anti-apoptotic role of polyphenols [38,40]. Subsequently, we studied if there were differences in the proliferative levels in the tissue from all experimental groups, due to the regenerative capacity of the liver in response to injury. However, we found that the positive nuclear immunoreactivity was limited to Kupffer cells in the liver sections from MS animals and that the administration of RSV plus QRC decreased the levels of proliferation (Figure 7). These results suggest that the pathway through which the RSV + QRC treatment reverses apoptosis and fibrosis generated by MS is related to the decreased expression of P2Y2 and TLR4 receptors thus diminishing inflammation. This is linked to the fact that there are cellular infiltrates and higher levels of NE. The differences that we found in the analysis of cellular proliferation with those of other studies could be due to the experimental model of MS and the tested doses of RSV + QRC used here as the regenerative response of the liver has been previously reported by other authors in experimental models of acute hepatic damage [41]. 

PPAR α and PPAR γ play a pivotal role in the control of several cardiometabolic diseases including liver diseases and these nuclear receptors bind FFA as their physiological ligands. Indeed, activated PPARs exert anti-inflammatory activities in several models through their ability to antagonize other signaling pathways [18]. They interact with other proteins, including Nuclear factor kappa B, Activated protein-1, and AMP-activated protein kinase, and they also downregulate TLR4 [20,42]. We found that livers from MS rats show a decreased expression of PPAR α and that PPAR γ is upregulated (Figure 8). These results suggest a relationship between the increase in inflammatory components, such as TLR4 and NE, and were consistent with other studies which indicated that PPARs receptors play a protective role in attenuating liver fibrosis [43,44,45]. Furthermore, there are conflicting reports on the contribution of PPAR α in various liver cell types to regulate cell proliferation [46,47]. Although the treatment with these concentrations of polyphenols did not affect the PPARs expression in MS animals, RSV + QRC could be regulating PPARs activity. To further clarify this point, it would be important to evaluate the effect of RSV + QRC administration on gene expression of transcriptional targets of PPARs in livers.

Finally, we analyzed the levels of transaminases to evaluate if the damage observed in the liver was reflected in their leak into the bloodstream. In this study we observed a significant increase in serum ALT and ALP activities in the MS rats when compared to Control rats and that the polyphenol mixture improved levels of these hepatic markers. These results are consistent with observations by other authors showing the hepatoprotective effects of natural compounds [4,39,48].

## 4. Materials and Methods

### 4.1. Animals and Surgical Procedures

All of the experiments were conducted in accordance with the ethical guidelines of the Instituto Nacional de Cardiología Ignacio Chávez (protocol #14-860). Male Wistar rats, 25 days old and weighing 45 ± 9 g, were randomly separated into two groups of 12 animals: group 1, Control rats that were given tap water for drinking, and group 2, MS rats that received 30% sugar in their drinking water during 20 weeks. Half of each group of rats (Control or MS) received their sucrose solution or drinking water with a mixture of RSV and QRC every day for four weeks in a dose 50–0.95 mg/kg/day, respectively (provided by ResVitalé^TM^, which contains 20 mg of QRC per 1050 mg of RSV). Groups without RSV + QRC treatment only received the vehicle (*n* = 6 per group). The mixture of RSV and QRC had been previously dissolved in 1 mL ethanolic solution (20%). The animals were maintained under standard conditions of light and temperature with water and food (LabDiet 5001; Richmond, IN, USA) ad libitum. At the end of the treatment, the animals were weighed and systolic arterial blood pressure was determined in conscious animals by a plethysmographic method previously described [22]. After overnight fasting, rats were euthanized by decapitation by guillotine. The intra-abdominal white adipose tissue (retroperitoneal fat pad) was carefully dissected with scissors, wet weight was determined, and then the tissue was discarded. The livers were excised and divided for histological analyses while fresh.

### 4.2. Measurement of Serum Biochemical Parameters

The fasting measurements of glucose, total cholesterol, and triglycerides were performed with commercial enzymatic kits (RANDOX Laboratories Ltd., Crumlin, Country Antrim, UK). Serum insulin levels were measured using a rat-specific insulin radioimmunoassay (Linco Research, Inc., Saint Charles, MO, USA). IR was estimated from the homeostasis model (HOMA-IR), as previously described [24].

Serum glutamic-oxaloacetic transaminase (SGOT/AST), glutamic pyruvic transaminase (SGPT/ALT), alkaline phosphatase (ALP), and γ-glutamyl transferase (GGT) activities were determinate spectrophotometrically using UV-test, International Federation of Clinical Chemistry [IFCC] (Roche Cobas C-501, Roche Diagnostics, IN, USA) [49].

### 4.3. Liver Tissue Preparation and Histological Examinations

The liver tissue of each group was processed to make frozen sections (10 μm). Sections for colorimetric staining (Picro-Sirius Red (SR)) and Masson’s trichrome (MT) were placed on gelatinized slides. The sections for purinergic receptor P2Y2, TLR4, and NE, and PCNA were placed on electro-charged slides. The photomicrographs for MT and SR were taken with a QIMAGING Micropuplisher 5.0 camera with Real-Time Viewing (RTV) coupled to an Olympus BX5 microscope. The images for PCNA were acquired with a Carl Zeiss microscope (Carl Zeiss Microscopy GmbH, Jena, Germany). Analysis and quantification of the area with collagen deposits (MT and SR) and with a signal for P2Y2, TLR4, NE, and the percentage of PCNA and TUNEL positive cells was performed with Image-Pro Premier Version 9.0 software (Media Cybernetics, Inc., Rockville, MD, USA). Four fields of each animal (*n* = 6) were analyzed, for a total of 16 fields of each condition in 20× photomicrographs. 

#### 4.3.1. Fibrosis Detection

For the detection of fibrosis, the staining was performed with Accustain Trichrome Stain (MT) Kit (Sigma-Aldrich, HT15) and Picro-SR solution (ab246832; Abcam PLC, Cambridge, UK) following the manufacturer’s instructions.

#### 4.3.2. Immunofluorescence

The sections were incubated in blocking solution for 1 h at room temperature. The incubation with the primary antibodies at a dilution of 1: 500 [(anti-P2Y2 (sc-518121], anti-TLR4 (sc-518121) and anti-NE (sc-55549) was carried out overnight at 4 °C. A 1:200 dilution of the secondary antibody m-IgGκBP (sc-516141) was used for NE and P2Y2; while for TLR4 a 1:400 dilution of mouse anti-rabbit IgG (sc-3753) with overnight incubation at 4 °C was done (all from Santa Cruz Biotechnology, CA, USA). Nuclei were labeled with DAPI. Observation and photographs for fluorescence images were obtained with a Cell Imaging Station (Life Technologies, Carlsbad, CA, USA).

#### 4.3.3. Apoptosis and Proliferation Analysis

The apoptosis of liver cells was detected using the In Situ Cell Death Detection Kit, TMR (tetramethylrhodamine-5-dUTP) red, version 12 (12156792910; Roche Applied Science, Mannheim, Germany) according to the manufacturer’s instructions. Sections were mounted with DAPI and observed in fluorescence microscopy (FLoid™ Cell Imaging Station). The percentage of TUNEL positive cells was calculated.

For PCNA, the incubation for 48 h at 4 °C with mouse monoclonal antibody (13-3900-Invitrogen Biotechnology, Waltham, MA, USA) (1:50) and the incubation for 1 h at 37 °C with m-IgGκ BP-HRP:sc-516102 (Santa Cruz Biotechnology, CA, USA) (1:500) as secondary antibody was carried out. The signal was revealed with the 3,3′-Diaminobenzidine (DAB)/chromogen substrate and hematoxylin. The images were captured in a Carl Zeiss microscope (Carl Zeiss Microscopy GmbH, Jena, Germany).

### 4.4. Western Blotting Analysis

The livers were homogenized in a lysis buffer pH = 8 (25 mM Hepes, 100 mM NaCl, 15 mM Imidazole, 10% glycerol, 1% Triton X-100) and protease inhibitor cocktail. The homogenate was centrifuged at 19,954× *g* for 10 min at 4 °C; the supernatant was separated and stored at −70 °C. The Bradford method was used to determine the total proteins [50].

A total of 50 μg protein was separated on an SDS-PAGE (12% bis-acrylamide-laemmli gel) and transferred to a polyvinylidene difluoride (PVDF) membrane. Blots were blocked for 1 h at room temperature using Tris-buffered saline (TBS)-0.01% Tween (TBS-T 0.01%) plus 5% non-fat milk. The membranes were incubated overnight at 4 °C with rabbit primary polyclonal antibodies PPAR-α, and PPAR-γ from Santa Cruz Biotechnology (Santa Cruz, CA, USA) as previously described [26]. All blots were incubated with Glyceraldehyde 3-phosphate dehydrogenase (GAPDH) antibody as a loading control. Images from films were digitally obtained by GS-800 densitometer with the Quantity One software (Bio-Rad Laboratories, Inc. Hercules, CA, USA) and they are reported as arbitrary units (AU).

### 4.5. Statistical Analysis

Results are expressed as mean ± standard error of the mean (SEM). Differences were considered statistically significant when *p* < 0.05. The different letters (a and b) in tables and figures indicate significant differences. We applied a one-way analysis of variance (ANOVA) followed by a Bonferroni post hoc test using the SigmaPlot program version 11 (Jandel Scientific, San Jose, CA, USA). 

## 5. Conclusions

The most important outcome of the present study was that there is a downregulation of the expression of TLR4, NE, and P2Y2. This is a new mechanism underlying the beneficial effects of RSV and QRC against inflammation in liver damage associated with MS. This effect leads to a decrease in apoptosis and fibrosis with no changes in hepatocytes proliferation. In addition, PPAR alpha and gamma expressions were altered in MS but their expression was not affected by the treatment with the natural compounds.

## Figures and Tables

**Figure 1 ijms-22-08939-f001:**
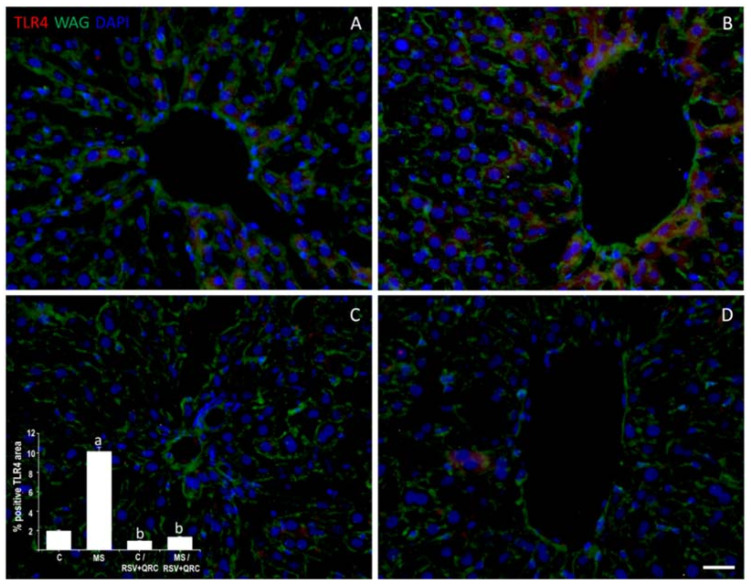
RSV + QRC administration decreased TLR4 expression in the liver from Control and MS rats. The detection of TLR4 immunostaining (red) was located in hepatocytes around of central vein. Wheat germ agglutinin (WAG) labeled with Oregon Green^®^ 488 was used to label the membranes and 2-[4-(Aminoiminomethyl) phenyl]-1H-Indole-6-carboximidamide hydrochloride (DAPI) for nuclei. The graph with the values of the percentage of positive TLR4 area is in the lower-left corner. ^a^ *p* < 0.05 vs. Control: ^b^ *p* < 0.01 vs. MS group. Panel (**A**) = Control group, Panel (**B**) = metabolic syndrome (MS) group, Panel (**C**) = Control/RSV + QRC group, Panel (**D**) = MS/RSV + QRC group. Bar = 50 μm.

**Figure 2 ijms-22-08939-f002:**
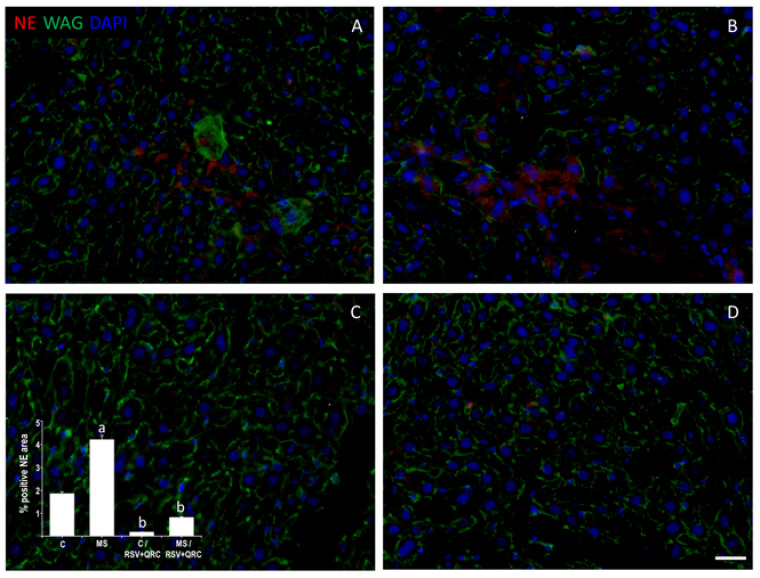
Effect of the administration of natural compounds on Neutrophil elastase (NE) immunodetection. The detection of NE immunostaining (red) was located in regions away from the lumen of the vessels. Wheat germ agglutinin (WAG) labeled with Oregon Green^®^ 488 was used to label the membranes and DAPI for nuclei. The graph with the values of the percentage of positive NE area is in the lower-left corner. ^a^ *p* < 0.05 vs. Control: ^b^ *p* < 0.01 vs. MS group. Panel (**A**) = Control group, Panel (**B**) = metabolic syndrome (MS) group, Panel (**C**) = Control/RSV + QRC group, Panel (**D**) = MS/RSV + QRC group. Bar = 50 μm.

**Figure 3 ijms-22-08939-f003:**
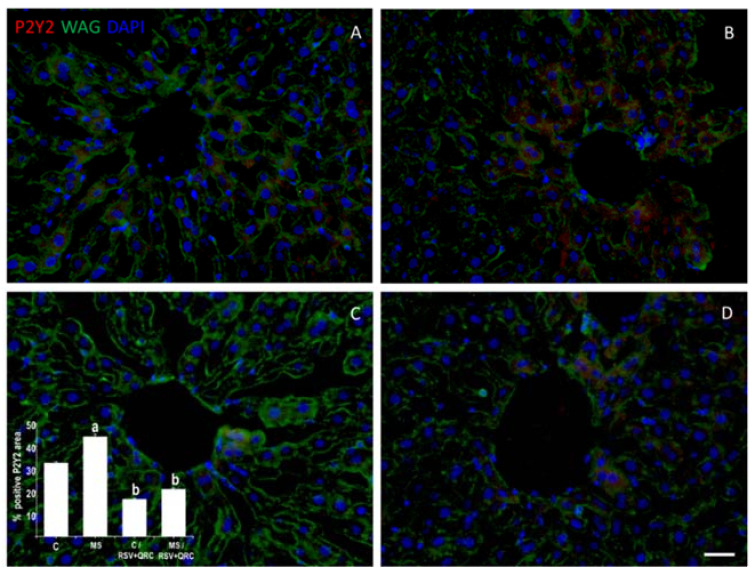
Effect of administration of RSV + QRC on the expression of the P2Y2 receptor in the liver from Control and MS rats. The detection of P2Y2 immunostaining (red) was located in hepatocytes around of central vein. Wheat germ agglutinin (WAG) labeled with Oregon Green^®^ 488 was used to label the membranes and DAPI for nuclei. The graph with the values of the percentage of positive P2Y2 area is in the lower-left corner. ^a^
*p* < 0.05 vs. Control: ^b^
*p* < 0.01 vs. MS group. Panel (**A**) = Control group, Panel (**B**) = metabolic syndrome (MS) group, Panel (**C**) = Control/RSV + QRC group, Panel (**D**) = MS/RSV + QRC group. Bar = 50 μm.

**Figure 4 ijms-22-08939-f004:**
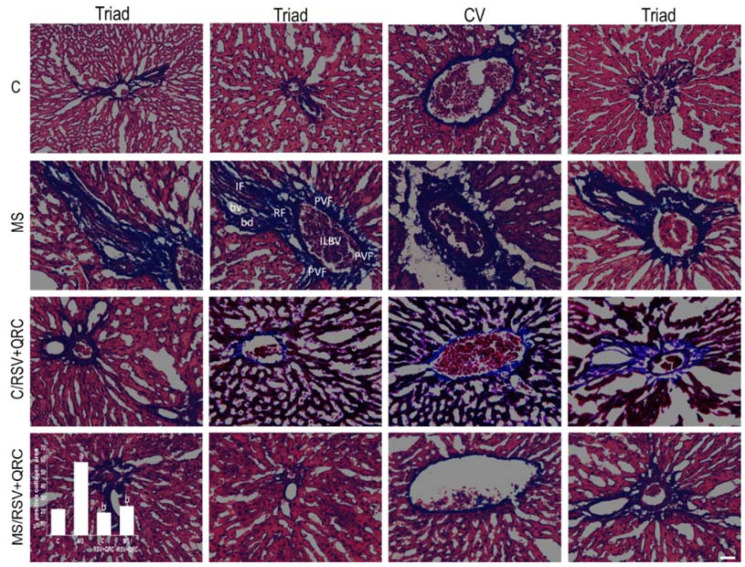
Resveratrol and quercetin administration attenuates liver fibrosis in the liver from MS rats. Representative images of Masson’s Trichrome staining; the triad and central vein (CV) in each condition are presented. The proportion of collagen deposits is greater in MS in the perivascular region (PVF), between (IF) and within the hepatocytes (RF) surrounding the vessels, both in the CV and in the triad. In the lower-left corner, the graph with the % total positive collagen area in each group is presented. ^a^ *p* < 0.05 vs. Control: ^b^
*p* < 0.01 vs. MS group. Abbreviations: C = Control, MS = metabolic syndrome; RSV + QRC = resveratrol plus quercetin, ILBV = interlobular vein, bd = bile duct, bv = blood vessel, PVF = perivascular fibrosis; IF = interstitial fibrosis, RF = replacement fibrosis. Bar = 50 µm.

**Figure 5 ijms-22-08939-f005:**
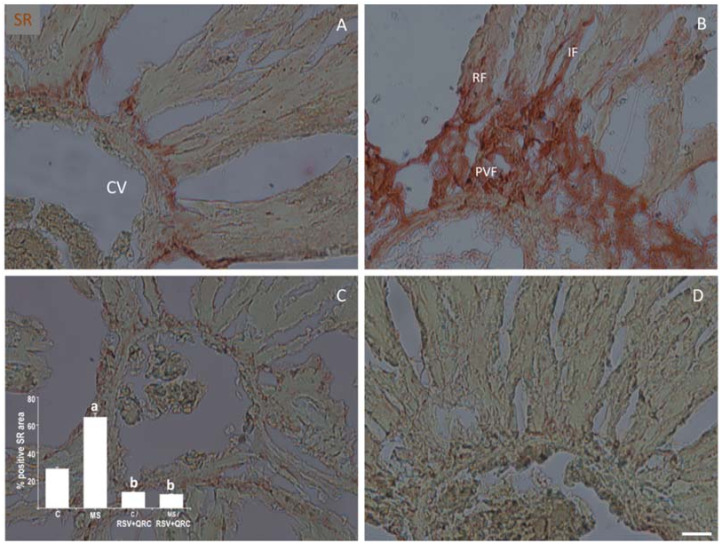
Effect of the administration of Resveratrol and Quercetin on fibrosis by Picrus-Sirius Red staining (SR) in livers from Control and MS rats. Representative images of the central vein (CV) of each condition are presented. Hepatocytes (RF) surrounding the vessels. The graph with the % SR positive (fibrosis) area in each group is shown in the lower-left corner. ^a^ *p* < 0.05 vs. Control: ^b^ *p* < 0.01 vs. MS group. Panel (**A**) = Control group, Panel (**B**) = metabolic syndrome (MS) group, Panel (**C**) = Control/RSV + QRC group, Panel (**D**) = MS/RSV + QRC group. Abbreviations: PVF = perivascular fibrosis; IF = interstitial fibrosis, RF = replacement fibrosis (hepatocytes surrounding the vessels). Bar = 50 μm.

**Figure 6 ijms-22-08939-f006:**
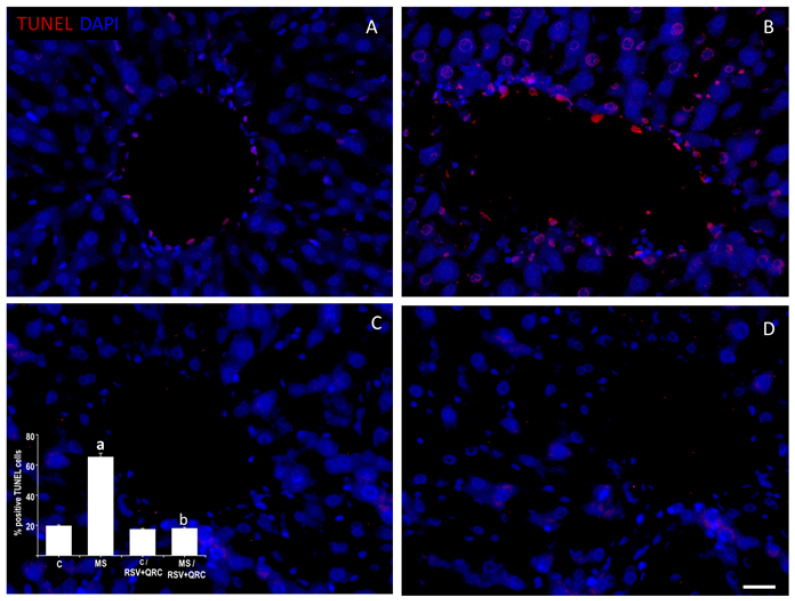
Resveratrol and quercetin treatment decreased apoptosis in livers from MS rats. The cells in apoptosis are located towards the lumen of the vessels as well as in the hepatocytes around the vessels, mainly in the central vein. The nuclei were marked with DAPI. The graph with the percentage of positive TUNEL positive cells is in the lower-left corner. ^a^ *p* < 0.05 vs. Control; ^b^ *p* < 0.01 vs. MS group. Panel (**A**) = Control group, Panel (**B**) = metabolic syndrome (MS) group, Panel (**C**) = Control/RSV + QRC group, Panel (**D**) = MS/RSV + QRC group. Bar = 50 μm.

**Figure 7 ijms-22-08939-f007:**
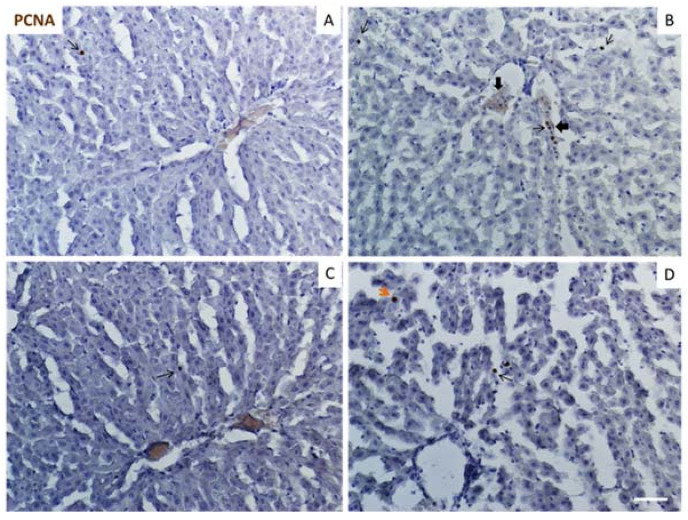
Expression of proliferating cell nuclear antigen (PCNA) in liver tissue from Control and MS rats. Thin arrows indicate the locations of PCNA positive cells. The thick arrows indicate cellular infiltrates and the orange arrow indicates a PCNA positive hepatocyte. Panel (**A**) = Control group, Panel (**B**) = metabolic syndrome (MS) group, Panel (**C**) = control/RSV + QRC group, Panel (**D**) = MS/RSV + QRC group. Bar = 100 μm.

**Figure 8 ijms-22-08939-f008:**
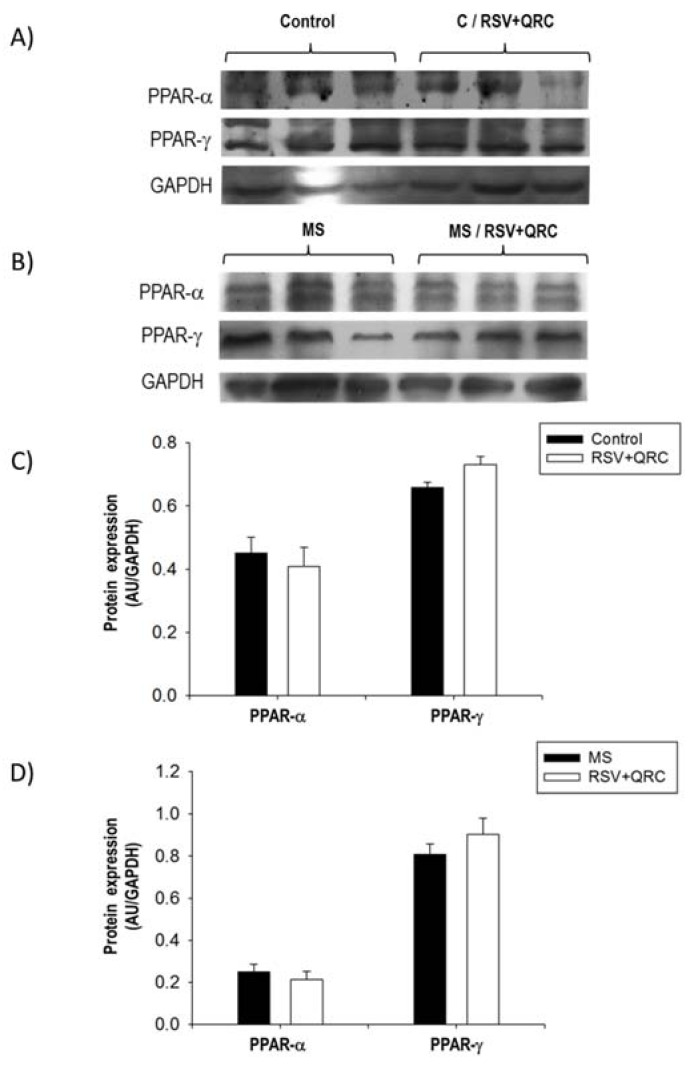
Effect of resveratrol and quercetin treatment on the expression of PPAR isotypes in the liver. Representative western blot from Control (**A**) and metabolic syndrome (MS) rats (**B**). (**C**) Expression of PPAR-α and PPAR-γ in Control group; (**D**) Expression of PPAR-αand PPAR-γ in MS group. Data represent mean ± SEM (*n* = 6 per group).

**Table 1 ijms-22-08939-t001:** The effects of the administration of RSV + QRC on body characteristics and biochemical parameters in Control and Metabolic syndrome (MS) rats.

	Control	Control/RSV + QRC	MS	MS/RSV + QRC
Weight (g)	492.7 ± 11.2	507.3 ± 15.7	583.3 ± 12.9 ^a^	441.3 ± 9.3 ^b^
Central adiposity (g)	5.2 ± 0.7	6.1 ± 0.9	13.1 ± 0.5 ^a^	8.1 ± 1.4 ^b^
Blood pressure (mm Hg)	101.7 ± 2.5	106.7 ± 2.4	143.6 ± 1.0 ^a^	113.2 ± 1.3 ^b^
Glucose (mg/dL)	92.1 ± 0.9	91.7 ± 0.8	96.5 ± 1.9	93.3 ± 1.0
Total Cholesterol (mg/dL)	56.2 ± 1.7	59.4 ± 3.2	64.1 ± 1.4	62.1 ± 0.8
Triglycerides (mg/dL)	83.6 ± 6.7	78.9 ± 4.2	145.2 ± 6.2 ^a^	98.3 ± 5.2 ^b^
Insulin (ng/mL)	0.15 ± 0.04	0.13 ± 0.02	0.48 ± 0.05 ^a^	0.17 ± 0.02 ^b^
HOMA index	0.91 ± 0.2	0.62 ± 0.12	2.41 ± 0.3 ^a^	0.7 ± 0.06 ^b^
Leptin (ng/dL)	2.6 ± 0.3	2.3 ± 0.1	5.3 ± 0.4 ^a^	4.1 ± 0.8
Adiponectin (μg/mL)	4.2 ± 0.3	3.8 ± 0.1	6.5 ± 0.5 ^a^	5.9 ± 0.3

Values are mean ± SEM. *n* = 6 in each group; ^a^
*p* < 0.01 MS without treatment vs. Control without treatment; ^b^ *p* < 0.01 vs. MS group. Abbreviations: MS, metabolic syndrome; HOMA-IR, Homeostatic model assessment of insulin resistance.

**Table 2 ijms-22-08939-t002:** Effects of polyphenols on liver function in experimental groups.

	Control	Control/RSV + QRC	MS	MS/RSV + QRC
ALT (U/L)	17.6 ± 3.2	21.0 ± 1.7	42.2 ± 9.1 ^a^	14.2 ± 2.5 ^b^
AST (U/L)	114.4 ± 8.4	87.2 ± 16.5	112.3 ± 15.6	103.7 ± 13.3
GGT (U/L)	4.8 ± 1.2	2.4 ± 1.5	4.8 ± 1.2	2.4 ± 1.5
ALP (U/L)	55.5 ± 3.7	60.0 ± 7.8	73.0 ± 5.4 ^a^	50 ± 4.0 ^b^

Values represented as means and standard errors, the different superscript letters mean a significant difference; ^a^ *p* < 0.05 vs. Control: ^b^ *p* < 0.01 vs. MS group. Abbreviations: MS: metabolic syndrome; ALT, alanine transaminase; AST, aspartate transaminase; GGT, gamma-glutamyltransferase: ALP, alkaline phosphatase.

## Data Availability

The data in our study are available from the corresponding author upon reasonable request.

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
