# Peer review of "Resveratrol and Quercetin as Regulators of Inflammatory and Purinergic Receptors to Attenuate Liver Damage Associated to Metabolic Syndrome"

_ijms, 2021, doi:10.3390/ijms22168939_

Round 1

Author Response

Dear editors,

Thank you for giving us the opportunity to submit a revised draft of the manuscript “Resveratrol and Quercetin as regulators of inflammatory and purinergic receptors to attenuate liver damage associated to metabolic syndrome. Role of PPARs” for publication in International Journal of Molecular Science. We appreciate the time and effort that you and the reviewers dedicated to providing feedback on our manuscript and we are grateful for the insightful comments on and valuable improvements to our paper. We have incorporated the suggestions made by the reviewers. Those changes are mark using track changes function within the manuscript. Please see below, in red, for a point-by-point response to the reviewer’ comments.

Best regards,

Ma. Esther Rubio-Ruiz, Ph. D.

Reviewer 1

The paper by Cano-Martínez et al. investigated the effects of Resveratrol and Quercetin relating to inflammation and purinergic pathway and the role of PPARs. Those two polyphenols were already utilized for the modulation of various disease-related pathways with great attention. The topic is interesting; however, the scientific writing and presentation of data from current research need significant improvements for the publication of the International Journal of Molecular Sciences.

Title: In lines 29-31, “Our results suggest that these effects were not associate with differences in the expression of PPAR alpha and gamma in the liver”, authors suggested that the effects of two polyphenols were not associated with PPARs. Then, emphasis should be taken on other pathways, not on PPARs. Consider changing the title to convey the main findings from the current study in a direct way.

R= According to reviewers suggestion, we modified the title by removing the final sentence.

Abstract

Line 19: ‘Liver disease’ – the phrase ‘liver disease’ encompasses a wide spectrum of liver diseases with various causes and symptoms. Please specify the type of liver disease if possible

R= The reviewer is correct in her/his suggestion. In our study, we refers to Nonalcoholic fatty liver disease (NAFLD).

  1. Line 30: were not associate → were not associated

R= We corrected the sentence.

Lines 24-32: These sentences should be revised to have a logical flow. List key findings from the current research and emphasize/summarize the key messages in the last 1 or 2 concluding sentences.

R= According the reviewer suggestion, we have reworked the final section of abstract.

Introduction

Line 37: (sucrose, fructose) → (sucrose and fructose)

R= We added the word “and”.

Lines 37-38: excessive sugar intake is not a single factor for the development of metabolic syndrome; therefore, the sentences should be revised considering other causes and risk factors of metabolic syndrome

R= The reviewer is correct in her/his suggestion. We added the additional factors for the development of metabolic syndrome (increase in the intake of sugars, lack of physical activity, and genetic predisposition) (Page 2, first parragrap).

Lines 41-42: reactive C protein (RCP) → C-reactive protein

R=We corrected the term.

Line 42: necrosis tumoral factor alpha (TNF-) → tumor necrosis factor-alpha (TNFalpha)

R= We corrected the term.

Lines 47-48: “Moreover, some reports sug-47 gest a link of liver inflammation and IR [4]” – avoid using ambiguous words such as ‘some’. Also, if authors use the word ‘some’, cite more references, or change the word into ‘singular word’.

R= The reviewer is correct in her/his suggestion. We added the references.

Line 58: Irs1 – write in “full name (abbreviation)” if the word is repeated; otherwise, write in the full name. In addition to this, many abbreviations were written without written in full name in advance. Please check and revise thoroughly.

R= The reviewer is correct in her/his suggestion. We write the full name,

We revised the text and wrote the full names of some abbreviations.

Line 85-86: cite references. Also, several sentences were written without citing relevant references. Please check and revise thoroughly.

R= We revised the text and added some references.

The reason for the selection of two polyphenols, resveratrol and quercetin, should be explained, not just stating that these compounds have several beneficial properties in various models.

R=According to reviewer suggestion, we added more information for the selection of resveratrol and quercetin in our study (page 2, last paragraph).

Results

Line 115: remove a period

R= We removed the period.

Lines 116 and 269: Figure 1 show → Figure 1 show(s or ed)

R= We added a “s”.

Line 224: Figure 5 → Figure 8

R= We corrected the number of the figure.

Figure 8(D) – When separate gels were used for different groups, recommend not comparing different groups in separate gels, such as comparing controls and MS/MS RSV QRC. Consider removing mark ‘a’ of statistical significance in panel (D).

R= The reviewer is correct in her/his suggestion. We removed the mark “a” in panel D.

Discussion

The link between tested mechanisms was not clearly explained. For example, alterations in purinergic receptors were significant; therefore, link these alterations with other pathways should be discussed with citing references.

Avoid restating results in a section of the discussion. Focus more on the deep discussion regarding current findings in comparison to the previous evidence.

Lines 351-354: the sentences were not appropriate at this position.

Several key components for the discussion section were missing; for example, integrative paragraphs of the key findings, strengths and limitations of the current study, suggestions and future directions, etc. Rewrite the section of the discussion thoroughly.

R= According to reviewer suggestion, we reworked most of the discussion.

Materials and Methods

The order of subsections in the current manuscript is not appropriate. There is a journal putting the section of Materials and Methods at the end of the main text without following the section of conclusions. Please refer to the other papers published in this journal and consider rearranging the section of 4. Materials and Methods.

R= The reviewer is correct in her/his suggestion. We corrected the order of this section.

Lines 357-358: write the name of the institution that approved the experiment protocol.

R=We added the name of our Institution.

Include details regarding how intra-abdominal fat was separated/collected, including types of fat depots collected.

R= The reviewer is correct in her/his suggestion. We collected the retroperitoneal fat pad which is located within the abdominal cavity along the dorsal wall of the abdomen behind the kidney but not attached to the kidney or mixed with perirenal brown fat. We included the details in 5.1 section.

The number of animals in groups was differently written in table and section of ‘4.1. Animals and Surgical Procedures’. Check the number of animals in each group, and write the correct number in the table and texts.

R= We corrected the number of animals used in our study.

Line 384: 10μ – check the unit

R= The correct unit is microns.

Line 385: (Picro-Sirius Red (SR)  → (Picro-Sirius Red (SR))

R= We corrected the term.

Lines 386-387: Neutro-386 phil elastase (NE) →NE

R= We corrected the term.

Abbreviations: check journal guidelines for the abbreviation usage and revise accordingly. It just uses the acronym or abbreviation after the first full mention.

R=We check and revise the abbreviations.

Lines 388-389: 1) include supplier information for ‘QIM-388 AGINGMicropuplisher 5 camera’ and ‘Olympus BX5 microscope’, and 2) explain RTV.

R= The reviewer is correct in her/his suggestion. We corrected and completed the information

Lines 420-421: It is not repeatedly written when the details of the supplier information were already written.

R= The reviewer is correct in her/his suggestion. We corrected the sentence.

Line 433: as a load Control →as a loading control

R= We corrected the term.

Lines 438-439: The different letters (a and b) in tables 438 and figures indicate significant variations – the word ‘variations’ should be revised.

R= The reviewer is correct in her/his suggestion. We replaced the word “variations” for “differences”.

Line 439: write the type of t-test. Also, consider using ANOVA when appropriate.

R= The reviewer is correct in her/his suggestion. To compare the groups, we applied one-way analysis of variance (ANOVA) followed by Bonferroni post hoc test and we confirmed the statistical differences. The description was added in the methods section (page 15, last paragraph).

Lines 440-441: “The Materials and Meth-440 ods should be described with sufficient details to allow” – it seems a part of writing guideline from the template that journal provides.

R= We removed the sentence.

Line 445: apoptosis, hepatic fibrosis  apoptosis and hepatic fibrosis

R= We added “and”.

Conclusions

Avoid simply restating results. Rather, integrate the key findings and emphasize the originality and strengths of the current study.

R= According the reviewer suggestion, we have reworked the conclusions.

References

Update the references with the most recent/relevant ones if possible.

R= According the reviewer suggestion, we added some relevant references

Reviewer 2 Report

For this reviewer, this manuscript is very interesting, well written and the experimental study well designed.

In virtue of reported results, this reviewer suggests to modify the title of the paper leaving out the following sentence " role of PPARs", because it creates an expectation.

Their results suggest that the effects, induced upon the treatment with resveratrol and quercetin, were not associate with differences in the expression of PPAR alpha and gamma in the liver. In virtue of these, the authors should well explain it in the discussion section.

Author Response

Dear editors,

Thank you for giving us the opportunity to submit a revised draft of the manuscript “Resveratrol and Quercetin as regulators of inflammatory and purinergic receptors to attenuate liver damage associated to metabolic syndrome. Role of PPARs” for publication in International Journal of Molecular Science. We appreciate the time and effort that you and the reviewers dedicated to providing feedback on our manuscript and we are grateful for the insightful comments on and valuable improvements to our paper. We have incorporated the suggestions made by the reviewers. Those changes are mark using track changes function within the manuscript. Please see below, in red, for a point-by-point response to the reviewer’ comments.

Best regards,

Ma. Esther Rubio-Ruiz, Ph. D.

Reviewer 2

For this reviewer, this manuscript is very interesting, well written and the experimental study well designed.

In virtue of reported results, this reviewer suggests to modify the title of the paper leaving out the following sentence " role of PPARs", because it creates an expectation.

R= According to reviewer suggestion, we modified the title by removing the final sentence.

Their results suggest that the effects, induced upon the treatment with resveratrol and quercetin, were not associate with differences in the expression of PPAR alpha and gamma in the liver. In virtue of these, the authors should well explain it in the discussion section.

R= The manuscript was revised and we have reworked some section of discussion according to reviewer suggestion.

Round 2

Reviewer 1 Report

The manuscript is well revised accordingly.